# Prevalence and Clustering of Cardiovascular Risk Factors among Medical Staff in Northeast China

**DOI:** 10.3390/healthcare9091227

**Published:** 2021-09-17

**Authors:** Jianxing Yu, Huanhuan Jia, Zhou Zheng, Peng Cao, Xihe Yu

**Affiliations:** Social Medicine and Health Service Management, School of Public Health, Jilin University, Changchun 130021, China; yjxjlu@163.com (J.Y.); jhh_1994@163.com (H.J.); zhengzhou19@mails.jlu.edu.cn (Z.Z.); cppengcao@163.com (P.C.)

**Keywords:** cardiovascular diseases, medical staff, risk factors, clustering, prevalence

## Abstract

Background: The clustering of cardiovascular disease (CVD) risk factors has become a major public health challenge worldwide. Although many studies have investigated CVD risk factor clusters, little is known about their prevalence and clustering among medical staff in Northeast China. This study aimed to estimate the prevalence and clustering of CVD risk factors and to investigate the association between relevant characteristics and the clustering of CVD risk factors among medical staff in Northeast China. Methods: A cross-sectional survey of 3720 medical staff from 93 public hospitals in Jilin Province was used in this study. Categorical variables were presented as percentages and were compared using the *χ^2^* test. Multiple logistic regression analysis was used to evaluate the association between relevant characteristics and the clustering of CVD risk factors. Results: The prevalence of hypertension, diabetes, dyslipidemia, being overweight, smoking, and drinking were 10.54%, 3.79%, 17.15%, 39.84%, 9.87%, and 21.75%, respectively. Working in a general hospital, male, and age group 18–44 years were more likely to have 1, 2, and ≥3 CVD risk factors, compared with their counterparts. In particular, compared with being a doctor, being a nurse or medical technician was less likely to have 1, 2, and ≥3 CVD risk factors only in general hospitals. Conclusions: The findings suggest that medical staff of general hospitals, males, and older individuals have a high chance associated with CVD risk factor clustering and that more effective interventions should be undertaken to reduce the prevalence and clustering of CVD risk factors, especially among older male doctors who work in general hospitals.

## 1. Introduction

Cardiovascular disease (CVD) has become the primary cause of death in China and around the world [1,2], accounting for an estimated 17.9 million deaths globally in 2019, and more than three-quarters of these deaths occur in low- and middle-income countries [3]. Moreover, the prevalence of CVD is increasing in China; it killed nearly 4 million people in 2016 [4]. The increasing burden of CVD has become a major public health problem.

At present, most of the research involves the general population [5,6,7], while research on medical staff is almost entirely absent. Medical staff are essential to protect the health of the general population. Especially during the outbreak of the COVID-19 epidemic, medical staff were fighting on the front line against the epidemic and saving the lives of patients, but they were neglecting their own health. Studies have pointed out that during the COVID-19 epidemic, at least 62 medical workers in China participating in the anti-epidemic effort died on duty, including 23 cases (37.1%) due to an early lack of protection who died from COVID-19, 23 cases (37.1%) due to CVD, 6 cases (9.7%) of possible CVD, and 10 cases (16.1%) due to other reasons [8]. The number of deaths caused by CVD even exceeds the number caused by infection due to insufficient early protection.

Hypertension, diabetes, dyslipidemia, being overweight, smoking, and drinking are the main risk factors for CVD [9,10,11,12]. A considerable number of studies have pointed out that the occurrence and development of CVD can be reduced through appropriate management and control of these six risk factors [13,14,15]. In addition, clustering multiple risk factors in the same person significantly increases the risk of CVD compared with having only a single risk factor [6,15,16].

Due to the characteristics of medical jobs, such as shift work, inflexible working hours, extended working hours, and heavy workloads, medical staff face extreme stress, which not only impairs their health, but also reduces their productivity and prevents them from performing their work effectively in the workplace [17,18,19,20]. The purpose of this study was to investigate the exposure and clustering of CVD risk factors (hypertension, diabetes, dyslipidemia, being overweight, smoking, drinking) among medical staff in Northeast China, and to analyze the individual characteristics (e.g., gender, age, marriage, education, and occupation) affecting their clustering, to provide a scientific basis for the formulation of CVD prevention strategies and measures.

## 2. Materials and Methods

### 2.1. Study Population

A cross-sectional survey of medical staff was implemented in Jilin Province from 21 December 2020 to 10 January 2021. In this study, a public general hospital and a public traditional Chinese medical hospital were selected from each county, and 25% of the urban public hospitals were selected from each city in Jilin Province. In general, a total of 93 public hospitals were selected as research objects by a stratified sampling method, including 50 general hospitals and 43 traditional Chinese medical (TCM) hospitals. Through convenience sampling, 20 doctors, 10 nurses, and 10 medical technicians were selected from each hospital. The study participants were selected as medical staff between the ages of 18 and 60. The subjects were substituted if they did not wish to participate in the study. Finally, a total of 3720 medical staff from 93 public hospitals in Jilin Province took part in the study.

### 2.2. Ethics Statement

The Ethics Committee of the School of Public Health, Jilin University, reviewed and approved the study protocol (NO. 2019-12-03). Each participating medical worker signed an informed consent form prior to data collection.

### 2.3. Data Collection and Measurement

All data were collected through standard questionnaires to ensure consistency and accuracy. The questionnaire included basic demographic information (e.g., sex, age, marriage, education, and occupation), health-related behaviors (e.g., smoking and drinking), as well as physical measurements (e.g., height, weight, and hypertension) and laboratory tests (e.g., diabetes and dyslipidemia). Physical measurements and laboratory tests were based on the medical staff’s physical examination data in the last 2 months. In addition, to ensure the quality and integrity of the questionnaire, the survey supervisor conducted a second review of the submitted questionnaires on the same day to determine the validity of each answer.

### 2.4. Assessment Criteria

The six major CVD risk factors were clearly defined as follows: hypertension was defined as having been treated with antihypertensive medication within the past 2 weeks, and/or an average systolic blood pressure (SBP) ≥ 140 mmHg and/or an average diastolic blood pressure (DBP) ≥ 90 mmHg [21]. Diabetes was defined as having been treated with anti-diabetes medication (insulin or oral hypoglycemic agents) and/or fasting blood glucose (FPG) ≥ 7.0 mmol/L [22]. Dyslipidemia was defined as having been treated with antilipemic medication or having at least one of the following: low-density lipoprotein cholesterol (LDL-C) ≥ 4.14 mmol/L, high-density lipoprotein cholesterol (HDL-C) < 1.04 mmol/L, triglycerides (TG) ≥ 2.26 mmol/L, and total cholesterol (TC) ≥ 6.22 mmol/L [23]. Overweight was defined as a body mass index (BMI) ≥ 24.0 kg/m^2^ [24]. Smoking was defined as having smoked at least one cigarette daily continuously over the past 30 days or at least 18 packs in total each year [25]. Drinking was defined as an average alcohol consumption of at least one (women) or two (men) standard drinks per day over the last 30 days, and the total amount of alcohol intake was calculated as the number of standard drinks (10 g of pure ethanol per drink) [26]. 

### 2.5. Clustering of CVD Risk Factors

The clustering of CVD risk factors was assessed based on the presence of six major risk factors: hypertension, dyslipidemia, diabetes, being overweight, smoking, and drinking. If one medical staff had 0, 1, 2, ≥3 major risk factors (RFs), then RFs = 0, RFs = 1, RFs = 2, RFs ≥ 3, respectively.

### 2.6. Statistical Analyses

Data were analyzed using IBM SPSS 25.0 software (IBM Corporation, New York, NY, USA). Categorical variables were presented as percentages and were compared using the *χ^2^* test. Adjusted odds ratios (ORs) and 95% confidence intervals (CIs) were calculated by multiple logistic regression, and 95% confidence intervals (CIs) that did not include one revealed that they were statistically significant. Statistical significance was set at *p*-value < 0.05.

## 3. Results

As shown in Table 1, among a total of 3720 medical staff, 2000 (53.76%) medical staff worked at general hospitals, and 1720 (46.24%) medical staff worked at TCM hospitals. More than two-thirds (62.69%) of the medical staff were women, and 74.22% of the medical staff were in the 18–44 age group. Nearly four-fifths (79.44%) of the medical staff were married, 62.31% of the medical staff had an undergraduate education level, and half of the medical staff were doctors. In brief, by hospital category, there were significant differences between age and education (*p* < 0.05) but no difference for gender, marriage, and occupation (*p* > 0.05). 

Table 2 shows that the prevalence of hypertension, diabetes, dyslipidemia, being overweight, smoking, and drinking was 10.54%, 3.79%, 17.15%, 39.84%, 9.87%, and 21.75%, respectively. The prevalence of hypertension, diabetes, and dyslipidemia was higher in general hospitals than in TCM hospitals (*p* ≤ 0.05). In addition, the prevalence of the six risk factors differed significantly by gender, being higher in men than in women (*p* < 0.001), especially for being overweight, smoking, and drinking. Furthermore, except for smoking and drinking, the prevalence of hypertension, diabetes, dyslipidemia, and being overweight differed significantly by age and marriage (*p* < 0.001). Their prevalence was higher in the 45–60 age group than in the 18–44 age group, and their prevalence was the lowest in the unmarried group compared with the other marriage groups. Except for dyslipidemia and drinking, the prevalence of the other factors showed decreasing trends with education level (*p* < 0.05). However, the prevalence of the six risk factors was the highest in the doctor group (*p* < 0.001).

Table 3 shows that the prevalence of RFs = 0, RFs = 1, RFs = 2, and RFs ≥ 3 was 45.89%, 29.68%, 13.92%, and 10.51%, respectively. Overall, the number of CVD risk factors differed significantly by hospital category, gender, age, marriage, education, and occupation (*p* < 0.001). Working in a general hospital, male, 45–60 age group, postsecondary education, and being a doctor had a higher prevalence of RFs = 1, RFs = 2, and RFs ≥ 3. However, unmarried individuals had the lowest prevalence of RFs = 1, RFs = 2, and RFs ≥ 3 compared with married individuals.

The results of the multiple logistic regression analysis are shown in Table 4, in terms of the adjusted OR (95% CIs) of 1, 2, ≥3 CVD risk factors when having 0 CVD risk factors was set as the reference category. Staff working in a general hospital, men, and the 45–60 age group were more likely to have 1, 2, and ≥3 CVD risk factors than staff working in a TCM hospital, women, and the 18–44 age group (*p* < 0.05). In addition, married and other staff were also more likely to have 1, 2, and ≥3 CVD risk factors than unmarried staff (*p* < 0.05). Moreover, as the number of CVD risk factors increased, the adjusted OR (95% CIs) also increased. In contrast, the adjusted ORs (95% CIs) of 1 and 2 CVD risk factors with an undergraduate education were 0.80 (0.66, 0.97) and 0.74 (0.56, 0.98) compared with those with a postsecondary education, respectively (*p* < 0.05). Compared with being a doctor, the adjusted OR (95% CIs) of ≥3 CVD risk factors for nurses was 0.50 (0.30, 0.84), and the adjusted ORs (95% CIs) of 2 and ≥3 CVD risk factors for medical technicians were 0.72 (0.54, 0.97) and 0.70 (0.49, 0.98), respectively (*p* < 0.05).

Table 5 shows the multiple logistic analysis of the CVD risk factor clustering by hospital category. The 0 CVD risk factors were set as the reference category. The results show that men, 45–60 years old, and married were more likely to have 1, 2, and ≥3 CVD risk factors than women, 18–44 years old, and unmarried (*p* < 0.05). In addition, as the number of CVD risk factors increased, the adjusted OR (95% CIs) also increased. It should be noted that for TCM hospitals, the adjusted ORs (95% CIs) of 1, 2, and ≥3 CVD risk factors were not significant for education or occupation (*p* > 0.05). In contrast, for general hospitals, the adjusted ORs (95% CIs) of RFs = 1 and RFs = 2 for those with an undergraduate education were 0.65 (0.48, 0.89) and 0.51 (0.34, 0.77) compared with those with a postsecondary education, respectively (*p* < 0.05). Moreover, compared with being a doctor, being a nurse or medical technician was less likely to have 1, 2, and ≥3 CVD risk factors only in general hospitals (*p* < 0.05).

## 4. Discussion

With the development of China’s economy and changes in people’s lifestyles, the prevalence of CVD and its related risk factors in China has been increasing year by year [4,7,27]. However, people’s understanding of the disease is still insufficient, resulting in a continuous increase in the prevalence and mortality of CVD in China [28,29]. This is the first study to assess the prevalence and clustering of major CVD risk factors in a medical worker population in Northeast China.

This cross-sectional study was based on medical staff, and this study found that being overweight and alcohol consumption were the top two risk factors for CVD among medical staff. In addition, the prevalence of being overweight was higher than the average rate in the general adult population [5,6]. This finding may be due to Jilin Province being located in the central part of Northeast China, which has a temperate continental monsoon climate and an annual average temperature of 4.8℃. This climate leads people to eat a lot of meat and not engage in outdoor sports, especially in the cold winter [15]. Furthermore, according to the Global Burden of Disease study, the number of deaths attributable to alcohol consumption in China rose from 368,000 in 1990 to 70,300 in 2017 [30], and other studies have also pointed to the heavy economic burden of alcohol-related deaths in China [31,32,33]. However, the prevalence of dyslipidemia, hypertension, diabetes, and smoking were significantly lower than those found in other studies [34,35,36,37,38], which may be related to the medical occupation. Compared with the general population, medical staff know more about the prevention and control of related diseases and the harm of smoking on the body.

This study also found that the prevalence of hypertension, diabetes, and dyslipidemia was higher among the staff of general hospitals than TCM hospitals (*p* ≤ 0.05). At the same time, compared with TCM hospitals, general hospitals had a higher prevalence of risk factors 1, 2, and ≥3, which may be due to general hospital medical staff having more work stress and a higher workload because the number of patients treated in general hospitals is much higher than that in TCM hospitals. In addition, the prevalence of the risk factors differed significantly by gender, being more predominant among men (*p* < 0.001), especially for being overweight, smoking, and drinking. In addition, compared with women, men had a higher prevalence of RFs = 1, RFs = 2, and RFs ≥ 3, similar to the findings of other previous studies [27,39,40]. This result may be because men assume more responsibilities in society and tend to have more social parties, drink more alcohol, and smoke more cigarettes than women. In contrast, women tend to be more aware of their weight, especially during young and middle ages, which may translate into a favorable cardiovascular risk profile. Furthermore, except for smoking and drinking, the prevalence of hypertension, diabetes, dyslipidemia, and being overweight showed differences by age and marriage status (*p* < 0.001). The 45–60 age group had a higher prevalence of RFs = 1, RFs = 2, and RFs ≥ 3 than the 18–44 age group, which is similar to the findings of previous studies [5]. With increasing age, physical function declines, leading to a higher prevalence of hypertension, diabetes, dyslipidemia, and being overweight than the younger population, while smoking and drinking alcohol are personal habits that are not affected by age. In addition, unmarried individuals had the lowest prevalence of RFs = 1, RFs = 2, and RFs ≥ 3 than married individuals, possibly because unmarried people in general are younger and under less pressure. Except for drinking, the prevalence of risk factors was higher among those with postsecondary education (*p* < 0.05). In addition, compared with the other groups, those with postsecondary education had the highest prevalence of RFs = 1, RFs = 2, and RFs ≥ 3, which may be related to a higher education level and a better awareness of disease prevention and control [15]. Moreover, the prevalence of the six risk factors was the highest in the doctor group (*p* < 0.001). Compared with other groups, doctors had a higher prevalence of 1, 2, and ≥3 risk factors. Other studies have also pointed out that doctors have the most work stress and the highest workloads [41,42,43].

In addition, this study found that individuals working in a general hospital, men, and the age group 18–44 were more likely to have 1, 2, and ≥3 CVD risk factors, compared with their counterparts. Furthermore, the adjusted ORs were lower than those in other studies [5,6], possibly because the study subjects were medical staff. Medical staff generally perform better regarding disease control and prevention than the general population. Finally, the clustering of CVD risk factors in different hospital categories was studied separately, which was similar to the overall study results. However, compared with being a doctor, nurses or medical technicians were less likely to have 1, 2, and ≥3 CVD risk factors only in general hospitals. Another study has also pointed out that doctors in general hospitals not only treat more patients with more complex conditions but also have greater work pressure and workloads than those in TCM hospitals [44]. Thus, doctors in general hospitals are more likely to have clustered CVD risk factors than nurses and medical technicians.

This study has the following limitations. First, the smoking and drinking status of the medical staff is based on self-reporting, which may have a certain reporting bias. Second, this study was a cross-sectional study, and it was not possible to determine the causal relationship between relevant characteristics and the clustering of CVD risk factors. Third, some other confounding factors that might have impacts on the clustering of CVD risk factors, such as socioeconomic factors, lifestyle (eating, physical activity), and work conditions (shift work, work hours), were not under consideration, which might be the limitation of our study. 

## 5. Conclusions

This cross-sectional study provides information on the regional prevalence and clustering of CVD risk factors among medical staff in Northeast China and fills an information gap. The findings suggest that individuals working in general hospitals, men, and older individuals have a high chance associated with CVD risk factor clustering and that more effective interventions should be implemented to reduce the prevalence and clustering of CVD risk factors, especially among older male doctors working in general hospitals.

## Figures and Tables

**Table 1 healthcare-09-01227-t001:** Descriptive characteristics of medical staff by hospital category.

Category	Subcategory	Total (*n* = 3720)	General Hospital (*n* = 2000)	TCM Hospital (*n* = 1720)	*χ^2^*	*p*
Gender	Man	1388 (37.31%)	769 (38.45%)	619 (35.99%)	2.40	0.12
	Woman	2332 (62.69%)	1231 (61.55%)	1101 (64.01%)		
Age	18–44	2761 (74.22%)	1453 (72.65%)	1308 (76.05%)	5.58	0.02
	45–60	959 (25.78%)	547 (27.35%)	412 (23.95%)		
Marriage	Unmarried	638 (17.15%)	319 (15.95%)	319 (18.55%)	5.69	0.06
	Married	2955 (79.44%)	1618 (80.9%)	1337 (77.73%)		
	Other	127 (3.41%)	63 (3.15%)	64 (3.72%)		
Education	Post-secondary Education	1083 (29.11%)	449 (22.45%)	634 (36.86%)	94.60	<0.001
	Undergraduate	2318 (62.31%)	1353 (67.65%)	965 (56.1%)		
	Postgraduate	319 (8.58%)	198 (9.9%)	121 (7.03%)		
Occupation	Doctor	1860 (50%)	1000 (50%)	860 (50%)	0.00	1
	Nurse	930 (25%)	500 (25%)	430 (25%)		
	Medical Technician	930 (25%)	500 (25%)	430 (25%)		

**Table 2 healthcare-09-01227-t002:** The prevalence of CVD risk factors by relevant characteristics.

Category	Subcategory	Hypertension	Diabetes	Dyslipidemia	Overweight	Smoking	Drinking
Total	*N* (%)	392 (10.54%)	141 (3.79%)	638 (17.15%)	1482 (39.84%)	367 (9.87%)	809 (21.75%)
Hospital Category	General Hospital	229 (11.45%)	92 (4.60%)	397 (19.85%)	793 (39.65%)	199 (9.95%)	454 (22.70%)
	TCM Hospital	163 (9.48%)	49 (2.85%)	241 (14.01%)	689 (40.06%)	168 (9.77%)	355 (20.64%)
	*χ* ^2^	3.82	7.78	22.18	0.06	0.04	2.31
	*p*	0.05	<0.01	<0.001	0.80	0.85	0.13
Gender	Man	238 (17.15%)	88 (6.34%)	353 (25.43%)	852 (61.38%)	356 (25.65%)	581 (41.86%)
	Woman	154 (6.60%)	53 (2.27%)	285 (12.22%)	630 (27.02%)	11 (0.47%)	228 (9.78%)
	*χ* ^2^	102.60	39.47	106.88	428.80	620.24	526.24
	*p*	<0.001	<0.001	<0.001	<0.001	<0.001	<0.001
Age	18–44	163 (5.90%)	49 (1.77%)	342 (12.39%)	1006 (36.44%)	262 (9.49%)	596 (21.59%)
	45–60	229 (23.88%)	92 (9.59%)	296 (30.87%)	476 (49.64%)	105 (10.95%)	213 (22.21%)
	*χ* ^2^	243.96	119.32	171.05	51.74	1.71	0.16
	*p*	<0.001	<0.001	<0.001	<0.001	0.19	0.69
Marriage	Unmarried	11 (1.72%)	7 (1.10%)	43 (6.74%)	198 (31.03%)	59 (9.25%)	124 (19.44%)
	Married	360 (12.18%)	127 (4.30%)	569 (19.26%)	1224 (41.42%)	292 (9.88%)	660 (22.34%)
	Other	21 (16.54%)	7 (5.51%)	26 (20.47%)	60 (47.24%)	16 (12.60%)	25 (19.69%)
	*χ* ^2^	65.90	15.81	58.87	26.63	1.34	2.92
	*p*	<0.001	<0.001	<0.001	<0.001	0.51	0.23
Education	Post-secondary Education	156 (14.40%)	62 (5.72%)	198 (18.28%)	487 (44.97%)	131 (12.1%)	233 (21.51%)
	Undergraduate	213 (9.19%)	75 (3.24%)	385 (16.61%)	885 (38.18%)	208 (8.97%)	472 (20.36%)
	Postgraduate	23 (7.21%)	4 (1.25%)	55 (17.24%)	110 (34.48%)	28 (8.78%)	104 (32.6%)
	*χ* ^2^	25.40	18.70	1.46	18.37	8.56	24.73
	*p*	<0.001	<0.001	0.48	<0.001	0.01	<0.001
Occupation	Doctor	258 (13.87%)	99 (5.32%)	435 (23.39%)	848 (45.59%)	253 (13.60%)	504 (27.10%)
	Nurse	50 (5.38%)	15 (1.61%)	96 (10.32%)	243 (26.13%)	13 (1.40%)	103 (11.08%)
	Medical Technician	84 (9.03%)	27 (2.90%)	107 (11.51%)	391 (42.04%)	101 (10.86%)	202 (21.72%)
	*χ* ^2^	50.44	26.07	102.29	100.50	105.23	93.52
	*p*	<0.001	<0.001	<0.001	<0.001	<0.001	<0.001

**Table 3 healthcare-09-01227-t003:** The prevalence with different numbers of CVD risk factors.

Category	Subcategory	RFs = 0	RFs = 1	RFs = 2	RFs ≥ 3	*χ* ^2^	*p*
Total	*N* (%)	1707 (45.89%)	1104 (29.68%)	518 (13.92%)	391 (10.51%)		
Hospital Category	General Hospital	819 (40.95%)	596 (29.80%)	312 (15.60%)	273 (13.65%)	72.27	<0.001
	TCM Hospital	888 (51.63%)	508 (29.53%)	206 (11.98%)	118 (6.86%)		
Gender	Man	265 (19.09%)	484 (34.87%)	326 (23.49%)	313 (22.55%)	817.29	<0.001
	Woman	1442 (61.84%)	620 (26.59%)	192 (8.23%)	78 (3.34%)		
Age	18–44	1419 (51.39%)	813 (29.45%)	330 (11.95%)	199 (7.21%)	212.09	<0.001
	45–60	288 (30.03%)	291 (30.34%)	188 (19.60%)	192 (20.02%)		
Marriage	Unmarried	372 (58.31%)	183 (28.68%)	58 (9.09%)	25 (3.92%)	73.43	<0.001
	Married	1286 (43.52%)	879 (29.75%)	443 (14.99%)	347 (11.74%)		
	Other *	49 (38.58%)	42 (33.07%)	17 (13.39%)	19 (14.96%)		
Education	Post-secondary Education	432 (39.89%)	340 (31.39%)	174 (16.07%)	137 (12.65%)	28.13	<0.001
	Undergraduate	1128 (48.66%)	664 (28.65%)	297 (12.81%)	229 (9.88%)		
	Postgraduate	147 (46.08%)	100 (31.35%)	47 (14.73%)	25 (7.84%)		
Occupation	Doctor	681 (36.61%)	582 (31.29%)	318 (17.1%)	279 (15%)		
	Nurse	592 (63.66%)	232 (24.95%)	80 (8.60%)	26 (2.80%)	231.13	<0.001
	Medical Technician	434 (46.67%)	290 (31.18%)	120 (12.90%)	86 (9.25%)		

* “Other” included divorced and widowed.

**Table 4 healthcare-09-01227-t004:** The multiple logistic analysis of the CVD risk factor clustering.

Category	Subcategory	The Number of CVD Risk Factors and Adjusted OR (95%CIs)		
		RFs = 1	RFs = 2	RFs ≥ 3
Hospital Category	TCM Hospital	1	1	1
	General Hospital	1.43 (1.21, 1.69)	1.95 (1.54, 2.48)	3.09 (2.26, 4.22)
Gender	Woman	1	1	1
	Man	4.22 (3.47, 5.14)	9.77 (7.51, 12.71)	21.87 (15.49, 30.88)
Age	18–44	1	1	1
	45–60	1.58 (1.29, 1.95)	2.58 (1.96, 3.39)	3.86 (2.77, 5.38)
Marriage	Unmarried	1	1	1
	Married	1.48 (1.19, 1.84)	2.11 (1.49, 2.98)	4.07 (2.46, 6.73)
	Other *	1.83 (1.13, 2.96)	2.35 (1.13, 4.88)	7.29 (2.87, 18.48)
Education	Post-secondary Education	1	1	1
	Undergraduate	0.80 (0.66, 0.97)	0.74 (0.56, 0.98)	0.79 (0.56, 1.12)
	Postgraduate	0.79 (0.58, 1.10)	0.76 (0.48, 1.20)	0.60 (0.33, 1.1)
Occupation	Doctor	1	1	1
	Nurse	0.82 (0.67, 1.02)	0.84 (0.61, 1.17)	0.50 (0.30, 0.84)
	Medical Technician	0.95 (0.77, 1.16)	0.72 (0.54, 0.97)	0.70 (0.49, 0.98)

* “Other” included divorced and widowed. A multiple logistic regression model was used to estimate OR with 95% CIs, and all other factors were adjusted when OR with 95% CIs of each variable were estimated.

**Table 5 healthcare-09-01227-t005:** The multiple logistic analysis of the CVD risk factor clustering by hospital category.

Category	Subcategory	RFs = 1		RFs = 2		RFs ≥ 3	
		TCM Hospital	General Hospital	TCM Hospital	General Hospital	TCM Hospital	General Hospital
Gender	Woman	1	1	1	1	1	1
	Man	4.54 (3.45, 5.98)	3.93 (2.95, 5.23)	9.95 (6.69, 14.79)	9.53 (6.67, 13.61)	17.75 (10.20, 30.90)	24.09 (15.42, 37.64)
Age	18–44	1	1	1	1	1	1
	45–60	1.52 (1.13, 2.05)	1.65 (1.24, 2.21)	2.80 (1.86, 4.22)	2.45 (1.69, 3.55)	2.96 (1.77, 4.95)	4.62 (2.97, 7.19)
Marriage	Unmarried	1	1	1	1	1	1
	Married	1.37 (1.01, 1.86)	1.59 (1.16, 2.19)	2.17 (1.28, 3.69)	2.19 (1.37, 3.50)	5.30 (2.01, 13.99)	3.74 (2.03, 6.87)
	Other *	1.87 (1.03, 3.59)	1.71 (0.83, 3.53)	2.48 (0.87, 7.04)	2.26 (0.79, 6.44)	9.16 (2.05, 40.99)	7.04 (2.03, 24.41)
Education	Post-secondary Education	1	1	1	1	1	1
	Undergraduate	0.89 (0.69, 1.14)	0.65 (0.48, 0.89)	1.01 (0.7, 1.48)	0.51 (0.34, 0.77)	0.96 (0.58, 1.60)	0.67 (0.41, 1.10)
	Postgraduate	0.67 (0.40, 1.10)	0.79 (0.50, 1.23)	0.84 (0.4, 1.76)	0.61 (0.33, 1.11)	0.73 (0.27, 1.95)	0.51 (0.23, 1.14)
Occupation	Doctor	1	1	1	1	1	1
	Nurse	0.96 (0.70, 1.32)	0.71 (0.53, 0.95)	1.11 (0.67, 1.84)	0.68 (0.44, 0.99)	0.53 (0.21, 1.30)	0.49 (0.26, 0.90)
	Medical Technician	1.28 (0.96, 1.71)	0.68 (0.50, 0.91)	1.08 (0.70, 1.66)	0.50 (0.33, 0.74)	0.85 (0.47, 1.52)	0.60 (0.38, 0.98)

* “Other” included divorced and widowed. A multiple logistic regression model was used to estimate OR with 95% CIs, and all other factors were adjusted when OR with 95% CIs of each variable were estimated.

## Data Availability

Because of relevant regulations, the data cannot be shared.

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
