# Peer review of "Prevalence and Clustering of Cardiovascular Risk Factors among Medical Staff in Northeast China"

_healthcare, 2021, doi:10.3390/healthcare9091227_

Round 1
Reviewer 1 Report
- The title is confusing, there are no clustering methods in the paper at all.
- The statistical analysis including clustering methods should be done according to the title.
- Some of the tables should be shown as figures, as there overwhelm readers (tab. 2) and some of them should be in a supplement (tab. 3).
- The TCM group will not be comparable with other classical groups of medical staff in other works. I propose to skip it and show the relationships between CV diseases and socio-economical factors.
- What was the main idea to do the paper, as the results are well known and described in the literature?
Author Response
A reply to the comments of reviewer 1 on
“Prevalence and clustering of cardiovascular disease risk factors among Medical Staff in Northeast China”
Thank you very much for your interest in our studies and constructive comments on our manuscript. We have revised our manuscript following your valuable suggestions. Specific replies to your general comments are given below:
- The title is confusing, there are no clustering methods in the paper at all.
Response: Thank you for valuable comments and questions. The “clustering” does not refer to a specific statistical method, but rather to the clustering of many cardiovascular disease risk factors in one person. We apologize for not describing it clearly in this paper, and we have added some details about the “clustering”, please see Page 3 Line 106-110. Furthermore, there are many articles similar to our study, but with different populations (Khanal, M. K., et al. "Prevalence and clustering of cardiovascular disease risk factors in rural Nepalese population aged 40–80 years." BMC Public Health 18.1(2018). Xin, H., et al. "Prevalence and clustering of cardiovascular risk factors: a cross-sectional survey among Nanjing adults in China." Bmj Open 8.6(2018): e020530. Wu, D. M., et al.).
- The statistical analysis including clustering methods should be done according to the title.
Response: Thank you for your good suggestion and careful reminder. The “clustering” does not refer to a specific statistical method, but rather to the clustering of many cardiovascular disease risk factors in one person. We apologize for not describing it clearly in this paper, and we have added some details about the “clustering” in the Materials and Methods section. Please see Page 3 Line 106-110.
- Some of the tables should be shown as figures, as there overwhelm readers (tab. 2) and some of them should be in a supplement (tab. 3).
Response: Thank you very much for your kind attention. We also considered using Figures rather than Tables to present the results previously, but we found that the results presented by Figures were less clear and more chaotic than those presented by Tables. Besides, Table 2 not only shows that the prevalences of hypertension, diabetes, dyslipidemia, being overweight, smoking and drinking, but also shows that whether there were differences about the prevalence of the six risk factors in different categories. And other similar studies also present the same results as Table 2 (Khanal, M. K., et al. "Prevalence and clustering of cardiovascular disease risk factors in rural Nepalese population aged 40–80 years." BMC Public Health 18.1(2018). Xin, H., et al. "Prevalence and clustering of cardiovascular risk factors: a cross-sectional survey among Nanjing adults in China." Bmj Open 8.6(2018): e020530.), so that Table 2 can be better compared with them.
Table 3 shows the prevalence with different numbers of CVD risk factors (RFs), which also presented whether there were differences among all the group (RFs=0, RFs=1, RFs=2, RFs≥3) in different categories. The analysis result is to pave the way for the following multiple logistic analysis. After careful consideration, table 3 is appropriate in main document. Thank you again for your help.
- The TCM group will not be comparable with other classical groups of medical staff in other works. I propose to skip it and show the relationships between CV diseases and socio-economical factors.
Response: Thank you for your reminder. The object of our study is medical staff, and medical staff in traditional Chinese medicine (TCM) hospitals account for a large proportion in China. Besides, large hospitals for the treatment of various diseases are mainly divided into TCM hospital and general hospital in China. Further, investigating the exposure and clustering of cardiovascular disease risk factors (hypertension, diabetes, dyslipidemia, being overweight, smoking, drinking) among medical workers in different types of hospitals is almost entirely absent. Moreover, we wanted to research the impact of some individual characteristics (e.g. gender, age, marriage, education and occupation) on the clustering of cardiovascular risk factors, and interventions can be targeted at specific populations through the analysis of these individual factors. In the following study, we are going to look at some of the socioeconomic factors, lifestyle (eating, physical activity), and work conditions (shift work, work hours) to try to figure out what kind of interventions should be targeted at these specific populations. Socio-economical factors that might have impacts on clustering of CVD risk factors were not under consideration, which might be the limitation of our study. So we have added these on Page 11 Line 248-252..
- What was the main idea to do the paper, as the results are well known and described in the literature?
Response: Hypertension, diabetes, dyslipidemia, being overweight, smoking and drinking are the main risk factors for CVD. A considerable number of studies have pointed out that the occurrence and development of CVD can be reduced through appropriate management and control of these six risk factors. In addition, clustering multiple risk factors in the same person significantly increases the risk of CVD compared with having only a single risk factor. However, most previous studies have focused on what are the risk factors for cardiovascular disease without considering which factors affect these risk factors, and few studies have focused on medical staff. Based on this, the purpose of this study was to investigate the exposure and clustering of cardiovascular disease risk factors (hypertension, diabetes, dyslipidemia, being overweight, smoking, drinking) among medical staff in different types of hospitals in Northeast China, to analyse some individual characteristics (e.g. gender, age, marriage, education and occupation) affecting their clustering, and to provide a scientific basis for the formulation of cardiovascular disease prevention strategies and measures. Therefore, we think this study has some innovation and reference significance in the research ideas and objects.
We appreciate your insightful comments that greatly improved the quality and presentation of the manuscript. We hope the revised version is acceptable to Healthcare. Thank you again for your helpful and constructive comments.

Reviewer 2 Report
The study is interesting, but it needs methodological adjustments, which I present in detail below.
- Abstract:
1) It is a cross-sectional study. So, the use of the terms "risck factors" or "protective factors" is no appropriated, because OR is measurement of chance. You can use "high prevalence", "low prevalence", "high chance", "low chance".
2) You should present the prevalence of the clustering CV risk factors (1, 2, 3 or more).
- Methods:
1) You wrote that "the subject were excluded if they did not wish to participate in the study". But, if I multiply 40 (individuals) x 93 (hospitals) = 3,720 medical staff. Thus, how many people were excluded in this study? Is there replacement of sample lost?
2) You should present how you created the clustering of CV risk factors, because it was your main outcome variable.
3) You presented the clustering of CV risk factors in tables 3, 4 and 5, and you did not explain how you created the variable in the methods section.
4) You must improve the statistical analyses description. For example, you used multinomial logistical regression. Why? Have your main outcome 3 or more categories? You have to especify how you treated your main outcome.
5) I consider unecessary to present the table 5. The main results of table 5 have already presented in table 4.
- Discussion:
1) You can't discuss about the factors associated with the higher or lower prevalence of individuals CV risk factors, because the data were not multiple adjusted.
- Other issues:
1) You use a language like your study had longitudinal design. You have to adapt the language to a cross-sectional study.
2) There are a lot of residual confounding factors which you did not analyses. For example, lifestyle (eating, physical activity), work conditions (shift work, work hours), etc. Could you present these data in a new version of the manuscript?
Author Response
A reply to the comments of reviewer 3 on
“Prevalence and clustering of cardiovascular disease risk factors among Medical Staff in Northeast China”
Thank you very much for your helpful and constructive comments on our manuscript. We have revised our manuscript following your valuable suggestions. If there are other questions about the paper, we hope you could not hesitate to point out and to help us improve the quality of my paper. The following is the point-by-point responses to your comments.
- Abstract:
1) It is a cross-sectional study. So, the use of the terms "risck factors" or "protective factors" is no appropriated, because OR is measurement of chance. You can use "high prevalence", "low prevalence", "high chance", "low chance".
Response: Thank you for your good suggestion and careful reminder. We totally agree. We have revised it in the paper, please see on Page 1 Line 21-24, 25-26, and Page 11 Line 232-233, line 257.
2) You should present the prevalence of the clustering CV risk factors (1, 2, 3 or more).
Response: According to your suggestion, we have modified this part as " Table 3 shows that the prevalences of RFs=0, RFs=1, RFs=2, RFs≥3 were 45.89%, 29.68%, 13.92% and 10.51%, respectively." Please see detail on Page 3 Line 140-143.
- Methods:
1) You wrote that "the subject were excluded if they did not wish to participate in the study". But, if I multiply 40 (individuals) x 93 (hospitals) = 3,720 medical staff. Thus, how many people were excluded in this study? Is there replacement of sample lost?
Response: Thank you very much for your kind attention. In the course of our investigation, if the medical staff does not want to participate in the study, we will find another medical staff replacement. The sentence was changed as “the subject were substituted if they did not wish to participate in the study” in the paper. Please see detail on Page 2 Line 73.
2) You should present how you created the clustering of CV risk factors, because it was your main outcome variable.
Response: Thank you for your good suggestion and careful reminder. We totally agree. As you suggested, we have added the clustering of CVD risk factors in the paper, please see on Page 3 Line 106-110.
3) You presented the clustering of CV risk factors in tables 3, 4 and 5, and you did not explain how you created the variable in the methods section.
Response: We are so sorry for neglecting this important information clearly in the previous manuscript. Thank you for your reminding, we have added the clustering of CVD risk factors in the paper, please see on Page 3 Line 106-110.
4) You must improve the statistical analyses description. For example, you used multinomial logistical regression. Why? Have your main outcome 3 or more categories? You have to especify how you treated your main outcome
Response: As you kindly pointed out above, we have checked the method description and results of the original paper, and this was a typing error. Multiple logistic regression was used in our analysis, but it was described as multinomial logistical regression in this paper. We are so sorry for that and we have revised it in this paper. Please see on Page 3 Line 115.
5) I consider unecessary to present the table 5. The main results of table 5 have already presented in table 4.
Response: Thank you very much for your kind attention. Table 4 shows the factors influencing the clustering of CVD risk factors among all medical staff, and medical staff working in a general hospital was more likely to have 1, 2, and ≥3 CVD risk factors than working in a TCM hospital. Further, table 5 shows the factors influencing the clustering of CVD risk factors in different hospital category, and being a nurse or medical technician was low chance associated with CVD risk factor clustering compare to doctor only in general hospitals. Table 5 is a further analysis of Table 4.
- Discussion:
1) You can't discuss about the factors associated with the higher or lower prevalence of individuals CV risk factors, because the data were not multiple adjusted.
Response: Thank you for your reminding. The results have been multiple adjusted in the paper. We are sorry for that the results were descripted in a wrong way and we have revised it in this paper. Besides, the adjusted variables have been presented as a footnote behind Table 4 and Table 5, please see on Page 8 Line 165-166, Page 9 Line 176-177. Thank you again for your help.
- Other issues:
1) You use a language like your study had longitudinal design. You have to adapt the language to a cross-sectional study.
Response: Thank you for your good suggestion. We totally agree. And we have revised the language to a cross-sectional study.
2) There are a lot of residual confounding factors which you did not analyses. For example, lifestyle (eating, physical activity), work conditions (shift work, work hours), etc. Could you present these data in a new version of the manuscript?
Response: Thank you for your good suggestion and careful reminder. lifestyle (eating, physical activity), and work conditions (shift work, work hours) that might have impacts on clustering of CVD risk factors were not under consideration, which might be the limitation of our study. So we have added these on Page 11 Line 248-252. We wanted to research the impact of some individual factors (e.g. gender, age, marriage, education and occupation) on the clustering of cardiovascular risk factors, and interventions can be targeted at specific populations through the analysis of these individual factors. In the following study, we are going to look at some socioeconomic factors, lifestyle (eating, physical activity), and work conditions (shift work, work hours) to try to figure out what kind of interventions should be targeted at these specific populations.
Thank you very much for your positive comments of our manuscript and giving a lot of valuable advice on our manuscript. We appreciate your insightful comments that greatly improved the quality and presentation of the manuscript. We hope the revised version is acceptable to Healthcare.

Round 2
Reviewer 1 Report
I have no other comments.
Reviewer 2 Report
The authors modified the manuscript incorporing my suggestions.